# OpenReview forum: "Refuse without Refusal: A Structural Analysis of Safety-Tuning Responses for Reducing False Refusals in Language Models"
_ICLR.cc/2026/Conference — Submitted to ICLR 2026_

### Official Review · Reviewer_EfgL · 2025-10-21

**Soundness:** 3
**Presentation:** 2
**Contribution:** 2
**Rating:** 4
**Confidence:** 3

**Summary:**

This paper tackles false refusals by decomposing safety-tuned responses into two components: (i) a boilerplate refusal statement and (ii) a rationale explaining the refusal. The key result is that the statement, not the rationale, drives over-refusal: removing or delaying the statement and training on rationales only markedly reduces false refusals on pseudo-harmful prompts while largely preserving safety on genuinely harmful inputs and core capabilities across multiple base models.

**Strengths:**

1. The decomposition into statement vs rationale is simple yet powerful. They show that a simple data-side correction, paired with in-context learning, it substantially reduces over-refusal.
2. The method is simple, easy to implement, and incurs minimal engineering overhead.

**Weaknesses:**

1. The core idea is intuitive and simple, yet too simple. Decomposing safety responses into a boilerplate refusal statement vs. an explanatory rationale and then removing the statement is essentially a data formatting/ablative supervision change rather than a new modeling principle, learning objective, or algorithm.
2. The contribution largely reads as data formatting for one failure mode rather than a broader safety-alignment advance. Conceptual novelty is modest without a new objective, theory, or model-side mechanism.
3. The study relies on a single LLM judge (Llama-3.3-70B-Instruct) with a custom prompt. Broader evaluation is encouraged (more powerful model like GPT 5). Moreover, experiments are limited to small base models, so scalability to larger instruction/RLHF-tuned systems remains untested.

Minor:

1. Can u analyze first-k tokens instead of first token?
1. Safety and over-refusal should be analyzed jointly: does a trade-off exist, or can the proposed method improve both?

**Questions:**

See Weaknesses.

---

> ### Author Response · Authors · 2025-11-20
> **Response to Reviewer EfgL (1/2)**
>
> We sincerely thank the reviewer for taking the time to provide a careful and detailed assessment of our work. We appreciate the constructive remarks and the specific points raised, which helped us better identify where additional clarification and analysis were needed. We have incorporated further experiments and explanations below, and we hope they address the reviewer’s concerns.
>
> &nbsp;
>
> **1. On the role of a simple structural decomposition in safety alignment.**
>
> > Decomposing safety responses into a boilerplate refusal statement vs. an explanatory rationale and then removing the statement is essentially a data formatting/ablative supervision change rather than a new modeling principle, learning objective, or algorithm.
> >
>
> Thank you for the thoughtful observation and for engaging carefully with the motivation behind our approach. We agree that the core idea is simple, and we view this intentional simplicity as a strength. Our goal was to design a method that remains straightforward while providing a clear analytical lens and a structured framework for examining how existing safety-alignment practices shape model behavior. This design allows the resulting behavioral differences to be examined in a clear, interpretable, and controllable manner, which we see as the primary strength of the approach.
>
> This simplicity also brings practical advantages. The method is easy to reproduce and can be directly applied within existing safety-alignment pipelines, while still effectively reducing false refusals in our experiments. Its ease of use allows it to complement ongoing alignment efforts and provide insights that may inform the design of future safety-supervision practices.
>
> &nbsp;
>
> **2. Clarifying the conceptual contribution.**
>
> > Conceptual novelty is modest without a new objective, theory, or model-side mechanism.
> >
>
> We appreciate the reviewer’s careful consideration of the conceptual scope of our contribution. Our work introduces a new perspective on constructing safety signals by decomposing responses into their constituent components, rather than applying a surface-level formatting change. This perspective allows us to analyze more clearly how each component contributes to model behavior, which helps identify a fundamental contributor to a known side effect of current safety alignment, broadens the understanding of false refusals, and clarifies how supervision design can support more reliable alignment.
>
> The broader relevance of this perspective is supported by in-context learning experiments, which suggest that the rationale-only benefit is not tied to a particular dataset or fine-tuning configuration but instead reflects properties of the supervision structure itself. Taken together, our framework clarifies why certain alignment practices lead to false refusals and indicates how supervision design can better balance safety and helpfulness.
>
> &nbsp;
>
> **3. Evaluation using GPT-5.1.**
>
> > Broader evaluation is encouraged (more powerful model like GPT 5)
> >
>
> Thank you for this helpful suggestion regarding broader evaluation. Following your recommendation, we additionally evaluated all four base models used in our study (Llama-3.1-8B, Mistral-7B-v0.3, Gemma-2-9B, and Qwen2.5-7B) using a more powerful external judge, GPT-5.1. We used the same evaluation protocol as in our main experiments. Across all models and benchmarks, GPT-5.1 produced judgments that closely match those of Llama-3.3-70B-Instruct, and the core trend reported in our paper remains unchanged. The results are summarized in the table below:
>
> | Llama-3.1-8B | Advbench | Malicious | XSTest-Safe | OKTest |
> | --- | --- | --- | --- | --- |
> | Statement-Only | 0.03 | 0.02 | 0.48 | 0.44 |
> | Rationale-Only | 0.00 | 0.02 | 0.70 | 0.64 |
> | Statement and Rationale | 0.02 | 0.0 | 0.38 | 0.42 |
> | Request-Specific | 0.03 | 0.01 | 0.77 | 0.75 |
> | Generic | 0.03 | 0.0 | 0.66 | 0.63 |
>
> | Mistral-7B-v0.3 | Advbench | Malicious | XSTest-Safe | OKTest |
> | --- | --- | --- | --- | --- |
> | Statement-Only | 0.04 | 0.01 | 0.52 | 0.59 |
> | Rationale-Only | 0.03 | 0.0 | 0.76 | 0.78 |
> | Statement and Rationale | 0.00 | 0.00 | 0.48 | 0.53 |
> | Request-Specific | 0.05 | 0.01 | 0.83 | 0.86 |
> | Generic | 0.03 | 0.0 | 0.70 | 0.81 |
>
> | Gemma-2-9B | Advbench | Malicious | XSTest-Safe | OKTest |
> | --- | --- | --- | --- | --- |
> | Statement-Only | 0.01 | 0.0 | 0.50 | 0.56 |
> | Rationale-Only | 0.0 | 0.01 | 0.66 | 0.66 |
> | Statement and Rationale | 0.0 | 0.01 | 0.44 | 0.51 |
> | Request-Specific | 0.0 | 0.02 | 0.72 | 0.71 |
> | Generic | 0.0 | 0.0 | 0.60 | 0.52 |
>
> | Qwen2.5-7B | Advbench | Malicious | XSTest-Safe | OKTest |
> | --- | --- | --- | --- | --- |
> | Statement-Only | 0.01 | 0.00 | 0.32 | 0.49 |
> | Rationale-Only | 0.01 | 0.01 | 0.66 | 0.76 |
> | Statement and Rationale | 0.02 | 0.01 | 0.39 | 0.41 |
> | Request-Specific | 0.03 | 0.02 | 0.71 | 0.81 |
> | Generic | 0.02 | 0.0 | 0.62 | 0.75 |

---

> ### Author Response · Authors · 2025-11-20
> **Response to Reviewer EfgL (2/2)**
>
> **4. Scalability to larger models.**
>
> > Moreover, experiments are limited to small base models, so scalability to larger instruction/RLHF-tuned systems remains untested.
> >
>
> We are grateful for the reviewer’s concern about scalability and for highlighting this important aspect. Our use of base models was intentional: as noted in Section 4 of the paper, our objective is to isolate dataset-side contributors to false refusals without the confounding effects introduced by prior instruction tuning or safety alignment. Applying our method to fully aligned instruction/RLHF models would make such causal analysis difficult. In addition, we did not include large instruction-tuned releases for two practical reasons:
>
> - **Lack of comparability:** Instruction/RLHF models rely on *undisclosed* safety datasets and pipelines, preventing clear attribution of behavioral changes to our decomposition.
> - **Resource constraints:** Models with fully reproducible training pipelines (e.g., OLMo, Tulu) require substantial compute to safety-tune from scratch, which is beyond our resources.
>
> For these reasons, we focus on open-source base models and lightweight supervision, enabling clear and reproducible evaluation of our data-level decomposition.
>
> Importantly, our findings do scale to larger base models: we additionally evaluated Llama-3.1-70B and Qwen2.5-72B and observed the same trends, indicating that our observations are not limited to smaller models. The resulting compliance rates are presented below.
>
> | Llama-3.1-70B | Statement-Only | Rationale-Only | Statement and Rationale |
> | --- | --- | --- | --- |
> | Advbench | 0.0 | 0.01 | 0.02 |
> | Malicious | 0.0 | 0.02 | 0.02 |
> | XSTest-Safe | 0.38 | 0.75 | 0.50 |
> | OKTest | 0.42 | 0.68 | 0.53 |
>
> | Qwen2.5-72B | Statement-Only | Rationale-Only | Statement and Rationale |
> | --- | --- | --- | --- |
> | Advbench | 0.0 | 0.02 | 0.01 |
> | Malicious | 0.0 | 0.03 | 0.02 |
> | XSTest-Safe | 0.53 | 0.79 | 0.42 |
> | OKTest | 0.60 | 0.74 | 0.54 |
>
> &nbsp;
>
> **5. Analyze first-k tokens.**
>
> Thank you for this valuable methodological suggestion. Following your recommendation, we extended our entropy analysis beyond the first token by computing, for each query, the cumulative entropy difference between the Statement and Rationale model and the Rationale-Only model over the first K generated tokens (K = 1–4), and reporting the mean of these summed differences for each dataset. **Negative values indicate higher entropy for the Rationale-Only model.**  The results are shown below.
>
> |  | K=1 | K=2 | K=3 | K=4 |
> | --- | --- | --- | --- | --- |
> | Advbench | -1.0673 | -0.7842 | -1.0401 | -1.1474 |
> | Alpaca | -0.1438 | -0.1620 | -0.1904 | -0.1834 |
> | XSTest-Safe | -0.5912 | -0.5716 | -0.8038 | -0.8294 |
>
> Across all datasets and all values of *K*, we observe the same trend as in the original K = 1 analysis:
>
> - the Statement and Rationale model shows **lower entropy** for harmful and pseudo-harmful queries and **higher entropy** for benign ones,
> - whereas the Rationale-Only model maintains **consistently higher entropy** across query types.
>
> This consistency across K supports our claim that the presence of a refusal statement makes the model’s initial generation more deterministic.
>
> Regarding token-level attribution for K steps, conducting a large-scale analysis would require manual inspection for each query and is not feasible within the rebuttal period. We will be happy to incorporate the extended K-step token-level attribution analysis in the camera-ready version.
>
> &nbsp;
>
> **6. Trade-off between safety and false refusal.**
>
> Thank you for raising this important point about jointly considering safety and false refusal. We agree that safety and false refusal should be examined jointly. As shown in Section 5 and Table 2 of the paper, all evaluated models achieve consistently high safety performance (above 0.9), and rationale-only supervision reduces false refusals while maintaining safety, occasionally with a slight decrease. We will clarify this joint consideration of safety and false refusal more clearly in the revised version.

---

> > ### Author Response · Authors · 2025-11-26
> > **Gentle Reminder**
> >
> > **Dear Reviewer EfgL,**
> >
> > We sincerely appreciate the time and effort you have devoted to reviewing our paper and providing valuable feedback. Your comments have been carefully considered, and we have endeavored to address all questions and concerns. It would be greatly appreciated if you could let us know whether our responses sufficiently clarify the points you raised.
> >
> > If any areas would benefit from further clarification or additional explanation, please do not hesitate to let us know. We would be happy to provide any further details you may require. Thank you once again for your thoughtful review and your time.

---

### Official Review · Reviewer_8bez · 2025-10-26

**Soundness:** 2
**Presentation:** 2
**Contribution:** 2
**Rating:** 2
**Confidence:** 3

**Summary:**

This work provides a valuable decomposition of safe responses by decoupling them into a refusal statementand a refusal rationale. Through a series of analyses, the authors convincingly demonstrate that the refusal statement's reliance on superficial cues is a key factor leading to erroneous over-refusal. This insight offers an important direction for constructing safety-alignment data in future research.

**Strengths:**

The paper's claim that the refusal statement's reliance on superficial cues leads to erroneous over-refusal is both insightful and intuitive. This point is convincingly substantiated by extensive experiments across four different models.

**Weaknesses:**

+ While the extensive experiments in this paper establish a correlation between the reliance on superficial cues in refusal statements and excessive refusal, they primarily demonstrate its role as a contributing factor rather than providing a definitive causal explanation. Further analysis would be needed to rule out alternative explanations and solidify this claim.
+ While the paper's observation suggests a viable path to mitigate over-refusal by having LLMs output only a request-specific rationale, this approach poses a pragmatic challenge regarding usability. In practice, for a response to be easily digestible, it is often preferable for an LLM to present a clear conclusion either before or after its detailed reasoning. Omitting a definitive refusal statement might compromise the clarity and directness expected in real-world applications.
+ While the paper's observation—that request-specific rationales are key to reducing erroneous refusals—is valuable, it introduces a significant practical challenge. The collection of high-quality training data containing such detailed, query-specific rationales is likely to be prohibitively expensive, posing a barrier to the widespread implementation of this solution.

**Questions:**

1. While I agree with the authors' intuition that refusal statements can over-rely on superficial cues, the evidence and analysis presented in the paper do not yet robustly establish this causal link. To strengthen this central claim, the authors should provide more compelling evidence, for instance, through a theoretical justification or more direct, extensive analytical experiments.

2. As noted earlier, while the request-specific rationale approach reduces erroneous over-refusal, it introduces challenges such as decreased readability and higher costs for data collection. The authors' discussion of these trade-offs appropriately addresses pragmatic challenges and is valuable for understanding how these findings might be translated into practice.

---

> ### Author Response · Authors · 2025-11-20
> **Response to Reviewer 8bez (1/2)**
>
> We sincerely appreciate the reviewer’s careful reading of the paper and the detailed feedback. Thank you for taking the time to articulate your concerns and for providing thoughtful suggestions that helped us better clarify several aspects of our work. We hope that the additional analyses and explanations below address the reviewer’s comments.
>
> &nbsp;
>
> **1. Additional causal evidence for the link between refusal statements and superficial cues.**
>
> > Further analysis would be needed to rule out alternative explanations and solidify this claim.
> >
>
> Thank you for raising this important point concerning the need for stronger causal evidence. In response, we conducted an additional experiment designed to more directly test whether the repeated co-occurrence of refusal statements and risky tokens induces false refusals at inference time.
>
> We identified five high-frequency risky tokens—**kill, money, drug, steal, shoot**—that commonly appear in harmful and pseudo-harmful benchmarks. For each keyword, we constructed:
>
> - **Training set**: Approximately 51 harmful queries containing the keyword, paired with **Rationale-Only** responses (to avoid confounding factors), yielding a total of 256 training examples.
> - **Evaluation set**: 50 pseudo-harmful queries per keyword (250 total).
>
> We then created five modified training datasets, each injecting a refusal statement *only* into responses to queries containing one of the five keywords (e.g., adding a refusal only when the query contained “drug”). Using these datasets, we trained six Llama-3.1-8B variants. At inference time, we measured how each model responded to pseudo-harmful queries corresponding to its keyword.
>
> - Results (Llama-3.1-8B)
>
>
>     |  | Overall (CR) | Relative Keyword Compliance Ratio | Drug Queries | Kill Queries | Money Queries | Steal Queries | Shoot Queries |
>     | --- | --- | --- | --- | --- | --- | --- | --- |
>     | Rationale-Only | 0.75 | - | 40 / 10  | 41 / 9 | 30 / 20 | 32 / 18 | 44 / 6 |
>     | Drug-Statement | 0.71 | 0.843 | **35 / 15** | 43 / 7 | 21 / 29 | 35 / 15 | 44 / 6 |
>     | Kill-Statement | 0.70 | 0.921 | 40 / 10 | **38 / 12** | 21 / 29 | 32 / 18 | 44 / 6 |
>     | Money-Statement | 0.71 | 0.857 | 43 / 7 | 40 / 10 | **18 / 32** | 32 / 18 | 44 / 6 |
>     | Steal-Statement | 0.71 | 0.814 | 43 / 7 | 42 / 8 | 21 / 29 | **26 / 24** | 45 / 5 |
>     | Shoot-Statement | 0.68 | 0.814 | 40 / 10 | 40 / 10 | 21 / 29 | 32 / 18 | **36 / 14** |
>
>     (Entries are shown as “# compliant / # refused”. Bolded entries indicates the keyword affected by the added statements. The *Relative Keyword Compliance Ratio* indicates the model’s compliance rate on pseudo-harmful queries containing that keyword, expressed relative to the average compliance rate of the other models. Each “X-Statement” model is trained on a dataset where a (keyword independent) refusal statement is added only to responses for queries containing keyword X.)
>
> - In summary, the Rationale-Only model achieves the highest overall compliance, consistent with our main claim. Across all modified models, adding refusal statements for a specific keyword reliably reduces compliance for pseudo-harmful queries containing that same keyword, despite all other conditions being held constant.
>
> This pattern indicates that repeated pairing of a refusal statement with a particular risky token during training leads the model to anchor on that token, producing false refusals even for benign queries. Because the datasets differ *only* in the presence or absence of these keyword-specific refusal statements, this experiment provides more direct causal evidence that boilerplate refusal statements can induce superficial-cue–based refusal behavior.
>
> We thank the reviewer again for prompting this deeper analysis; and we will incorporate these findings and clarifications into the revised version.

---

> ### Author Response · Authors · 2025-11-20
> **Response to Reviewer 8bez (2/2)**
>
> **2. Practical usability of rationale-only supervision.**
>
> > Omitting a definitive refusal statement might compromise the clarity and directness expected in real-world applications.
> >
>
> We are grateful for this practical and thoughtful observation regarding usability. We would like to clarify that, in our safety-response decomposition, the rationale already conveys the models’ stance by explaining the potential harms of the requested action, the consequences it may lead to, and suggesting that the model can assist if the user submits a safe query. Even without an explicit refusal statement, these elements generally make the model’s intent sufficiently clear.
>
> More importantly, our primary goal is to analyze dataset-side factors that contribute to false refusals, particularly the common pattern in which a boilerplate refusal statement precedes the rationale, and to propose directions for constructing more reliable safety supervision. If an explicit refusal statement is desired in certain deployment settings, our experiments also suggest a practical direction to consider. As shown in Section 5 (Table 2) of the paper, placing the refusal statement after the rationale modestly reduces false refusals compared to the standard statement-first format. Taken together, our decomposition provides guidance that help inform how safety supervision can be structured in different deployment scenarios.
>
> &nbsp;
>
> **3. Practical challenge of collecting request-specific rationales.**
>
> > The collection of high-quality training data containing such detailed, query-specific rationales is likely to be prohibitively expensive, posing a barrier to the widespread implementation of this solution.
> >
>
> We appreciate the reviewer’s careful consideration of the practical challenges involved. We would like to clarify that the request-specific condition in our study was introduced for analytical purposes. Together with the generic condition, it allows us to examine which characteristics of rationales, in contrast to refusal statements, contribute to distinguishing harmful from pseudo-harmful queries.
>
> The intention of this experiment is therefore to improve understanding of how different components of safety responses function, rather than to prescribe that request-specific rationales represent a indispensable component of safety data construction. We view these findings as offering considerations that may be useful when designing safety data, rather than specifying any particular data-collection requirement.
>
> Additionally, in settings where prompt–refusal response pairs are already available, it may be feasible to generate request-specific rationales using standard data-generation pipelines (e.g., LLMs). Providing a simple additional instruction to explicitly reference the input query enables the model to revise the existing response so that its query-relevant aspects are stated explicitly. We acknowledge that the notion of “cost” varies across applications, so we offer this only as an additional observation rather than as a claim about the overall data-collection burden.

---

> > ### Author Response · Authors · 2025-11-26
> > **Gentle Reminder**
> >
> > **Dear Reviewer 8bez,**
> >
> > We sincerely appreciate the time and effort you have devoted to reviewing our paper and providing valuable feedback. Your comments have been carefully considered, and we have endeavored to address all questions and concerns. It would be greatly appreciated if you could let us know whether our responses sufficiently clarify the points you raised.
> >
> > If any areas would benefit from further clarification or additional explanation, please do not hesitate to let us know. We would be happy to provide any further details you may require. Thank you once again for your thoughtful review and your time.

---

> ### Comment · Reviewer_8bez · 2025-11-26
> **Follow-Up Novelty Concerns**
>
> Thank you for the authors' detailed response. The reply further elaborates on the observation that frequent co-occurrence of specific tokens and refusal statements in the training data may lead LLMs to learn incorrect shortcuts, thereby potentially causing over-refusal. However, this does not fully address my concerns regarding the novelty of the work.
>
> Although the idea that refusal statements may form "safe shortcuts" and lead to erroneous rejections is important for the safety alignment of LLMs, the observation put forward in this paper appears rather intuitive. Moreover, it seems highly similar to the oral work presented at ICLR 2025 [1], as also noted in the authors' discussion with reviewer 513G. This raises further questions about the specific contribution of this paper: **what incremental value does it offer beyond [1]?** As a follow-up to [1], I believe a more meaningful contribution would involve an in-depth analysis of the underlying training mechanisms behind the emergence of "safe shortcuts," along with proposing actionable solutions. Yet, the authors’ response to my earlier concern about practical challenges states that the paper aims to "analyze dataset-side factors that contribute to false refusals, particularly the common pattern in which a boilerplate refusal statement precedes the rationale, and to propose directions for constructing more reliable safety supervision."
>
> Additionally, I am not convinced that the proposed "directions for constructing more reliable safety supervision" are sufficiently reliable. Aside from the previously mentioned issue of high data collection costs, the Safety Rationale itself may also develop shortcuts. For instance, if a model is aligned using a large volume of data stating that one should not "kill" humans or animals because it "harms their lives," could it incorrectly associate phrases like "kill time" with "harming life"? Similarly, if many examples state that "shooting" people or animals is illegal, might the model wrongly link "shoot a plan" (e.g., in a project context) with "illegal"? Such associations would be even harder to filter out or standardize compared to refusal statements. This is precisely why I was hoping to see a deeper analysis based on training mechanisms and corresponding algorithmic solutions.
>
> For the reasons outlined above, my persisting concerns lead me to uphold my current score.
>
> [1] Qi, Xiangyu, et al. "Safety alignment should be made more than just a few tokens deep." _arXiv preprint arXiv:2406.05946_ (2024).

---

> ### Author Response · Authors · 2025-12-02
> **Follow-Up Response to Reviewer 8bez (1/2)**
>
> We thank the reviewer for their continued feedback. We would like to offer further clarification regarding the remaining comments.
>
> &nbsp;
>
> > what incremental value does it offer beyond [1]?
> >
>
> We appreciate the reviewer identifying the connection to [1]. However, we respectfully clarify that our work addresses a *distinct problem statement* and proposes a structurally different approach. While [1] provide an important characterization of “shallow alignment” in safety robustness, our work investigates how conventional supervision structures contribute to utility degradation, specifically false refusals. The distinctions are as follows:
>
> &nbsp;
>
> **1. Phenomenon Discovery vs. Structural Causality Analysis**
>
> **[1]** (Phenomenon): They analyze the model’s generative distribution and observe that safety signals are highly concentrated in the initial tokens of the refusal prefix. Subsequently, they show that perturbing this prefix leads to safety failures (jailbreaking).
>
> **Ours** (Structural Causality): We study how this structural dependency affects semantic understanding. Through our analysis (Section 6 of the paper) and the keyword-specific experiment included in the rebuttal, we identify **which component** of the supervision signal and **how** it causally shapes the model’s behavior. Specifically, we show that the refusal statement drives the model to rely on superficial cues, resulting in semantic blindness in pseudo-harmful settings.
>
> **2. Divergent Problem Statements: Jailbreaking vs. False Refusal**
>
> **[1]** (Safety Robustness): They investigate how models can be coerced into producing unsafe outputs by jailbreaking. Their problem formulation centers on attacker-driven failures, where safety mechanisms collapse under additional manipulation.
>
> **Ours** (False Refusal): In contrast, we study false-refusal, where the model refuses legitimate and benign requests due to oversensitivity in the supervision signal. This phenomenon occurs under normal, unperturbed usage poses a usability challenge.
>
> **3. Solution Paradigm: Deepening vs. Supervision Decomposition**
>
> **[1]** (Deepening): They propose a defense mechanism that further fine-tunes already aligned models. Using data augmentation and a constrained optimization objective, their method deepens the refusal prefix and improves robustness to perturbations.
>
> **Ours** (Supervision Decomposition): We propose a Rationale-Only supervision strategy that removes reliance on refusal statements altogether, enabling “refusal without refusal.” Rather than strengthening the safety prefix, our approach revisits its structural role and demonstrates the benefits of removing it.
>
> In summary, [1] highlight vulnerabilities arising from shallow safety alignment that primarily shifts the model’s initial response distribution. In contrast, our work utilizes supervision decomposition to examine the impact of distinct factors, demonstrating that embedding refusal statements into supervision encourages models to rely on superficial cues, which in turn affects utility. We view the two lines of research as complementary, addressing different aspects of safety alignment.
>
> &nbsp;
>
> ---
> **references**
>
> [1] Xiangyu, Qi, et al. "Safety Alignment Should Be Made More Than Just a Few Tokens Deep." International Conference on Learning Representations (2025).

---

> ### Author Response · Authors · 2025-12-02
> **Follow-Up Response to Reviewer 8bez (2/2)**
>
> > Safety Rationale itself may also develop shortcuts.
> >
>
> We appreciate the reviewer’s thoughtful concern regarding whether the Safety Rationale itself could develop shortcuts. However, we clarify that Rationales fundamentally differ from Refusal Statements in their propensity to form such shortcuts due to their structural nature.
>
> **1. Structural Distinction: Input-Agnostic vs. Input-Dependent**
>
> The shortcut we focus on arises from **boilerplate** refusal statements, which are largely input-agnostic—similar phrases appear across many refusal scenarios. This repeated surface-level pattern encourages the model to anchor on superficial tokens and trigger refusal behavior with limited contextual processing.
>
> In contrast, rationales are **input-dependent**. Phrases such as “is illegal” or “causes harm” emerge from contextual semantic evaluation. To generate a rationale, the model processes how the verb (e.g., “kill”) interacts with its argument (e.g., “time” versus “human”). As shown in Figure 4 of the paper, this leads semantically relevant input tokens—not just the trigger word—to contribute to the model’s generated response.
>
> **2. Empirical Evidence**
>
> Our empirical findings do not support the hypothesis that rationales create similar shortcuts.
> - **Performance on High-Frequency Risky Tokens**: Our keyword-specific causal experiment (in Response to Reviewer 8bez (1/2)) naturally reflects a setting similar to the reviewer’s concern: risky tokens (e.g., “kill”, “shoot”) repeatedly co-occur with rationales. Even under this concentrated exposure, the Rationale-Only model maintained higher compliance on pseudo-harmful queries, indicating that it focused on semantic context rather than anchoring on the keyword.
> - **Benchmark Results:** The XSTest-Safe benchmark contains pseudo-harmful queries such as “kill a process” or “shoot a photo”. If rationales had induced a shortcut linking like refusal statements, the Rationale-Only model would have failed on these examples. Instead, it exhibited substantially fewer false refusals than models trained with refusal statements (Table 2 of the paper).
>
> **3. Conclusion**
>
> Taken together, these results suggest that the Safety Rationale does not function as a shortcut. Rather, it serves as a **context-sensitive supervisory signal**, encouraging the model to perform semantic understanding instead of relying on superficial lexical cues.

---

### Official Review · Reviewer_513G · 2025-10-29

**Soundness:** 4
**Presentation:** 3
**Contribution:** 3
**Rating:** 8
**Confidence:** 4

**Summary:**

This paper focuses on the problem of false refusals in LLMs. The authors investigate this through a data-centric framework, where they decompose a refusal response into a templated statement and a rationale that explains why the request is unsafe. Through controlled finetuning across multiple model families, they find that refusal statements increase false refusals by making models rely on superficial cues, whereas rational-only supervision reduces false refusals while preserving safety and general-task performance. They further demonstrate the applicability of rationale-only supervision to in-context learning and inference-time mitigation settings.

**Strengths:**

1. **Controlled experimental design.**
- The authors carefully and structurally decompose refusal statements and rationales, and the experiments are systematically replicated across multiple model families and benchmarks.

2. **Thorough analysis.**
- Beyond aggregating metrics, the authors incorporate (in Section 6) entropy measurements, token-level attribution, template substitution tests, and human evaluation. These provide a rigorous understanding of how refusal statements shape false refusal patterns and support their claims robustly.

3. **Conceptual contribution and practical applicability.**
- After establishing the scientific understanding of the utility of rational-only supervision through the data-centric framework, the authors further apply this insight in the applicable setting of in-context learning and combination with inference-time mitigation methods (Section 7). This makes the contribution not only theoretically elegant but also practically impactful for real deployment scenarios.

**Weaknesses:**

**The refusal statement styles are limited.**
- The study primarily relies on a fairly uniform, boilerplate refusal statement format, but the statement can be more nuanced and varied in tone, structure, and context. I wonder how robust are the findings under synthetic variations that introduce stylistic and structural diversity.

**Questions:**

**Improving LLM safety is beyond relying on training data alone**. Many works use additional safety mechanisms such as refusal tokens [1] and post-hoc classifiers [2].  It would be interesting to see how rationale-only supervision integrates with other common alignment techniques.

[1] Jain, Neel, et al. "Refusal tokens: A simple way to calibrate refusals in large language models." arXiv preprint arXiv:2412.06748 (2024).

[2] Han, Seungju, et al. "Wildguard: Open one-stop moderation tools for safety risks, jailbreaks, and refusals of llms." Advances in Neural Information Processing Systems 37 (2024): 8093-8131.

---

> ### Author Response · Authors · 2025-11-20
> **Response to Reviewer 513G (1/2)**
>
> We deeply appreciate the reviewer’s thoughtful and highly encouraging evaluation. Thank you for taking the time to engage closely with the paper and for offering constructive suggestions that helped us improve the clarity and completeness of our work. We hope that the additional analyses and explanations provided below are helpful in addressing the reviewer’s remaining points.
>
> &nbsp;
>
> **1. Results for various refusal statements.**
>
> > I wonder how robust are the findings under synthetic variations that introduce stylistic and structural diversity.
> >
>
> Thank you for raising this thoughtful point and for encouraging us to examine stylistic robustness more carefully. We would like to clarify that we did not intentionally constrain refusal statements to a uniform format. Rather, the limited stylistic variation reflects characteristics of current safety-alignment practice: existing safety-tuning datasets typically contain a small set of repeated refusal expressions, and, as noted in prior work [1], models trained under common alignment pipelines naturally converge to using a narrow set of refusal tokens. Our use of the term *“boilerplate” refusal statement* therefore refers to an empirical property of widely adopted datasets and alignment procedures, not a constraint imposed in our experimental setup.
>
> In response to the reviewer’s suggestion, we conducted an additional experiment to evaluate robustness under synthetic stylistic and structural variation. We created 15 refusal statement variants spanning diverse tones and forms (formal, concise, empathetic, principle-driven). Examples include:
>
> - “Regrettably, I cannot assist with this type of request.” (Formal / Direct)
> - “That's not something I can provide guidance on.” (Concise / Minimalist)
> - “I want to be helpful, but I need to avoid giving information like that.” (Warm / Empathetic)
> - “According to my safety guidelines, I must refrain from supplying details.” (Principle-driven)
>
> We replaced the original refusal statements in our dataset with these stylistic variants and re-evaluated the model. The results (compliance rates) of this stylistic-variation experiment are summarized in the tables below.
>
> | Llama-3.1-8B | Statement and Rationale (paper / varied) | Statement-Only (paper / varied) | Rationale-Only (paper) |
> | --- | --- | --- | --- |
> | Advbench | 0.02 / 0.02 | 0.03 / 0.03 | 0.02 |
> | Malicious | 0.02 / 0.02 | 0.02 / 0.01 | 0.06 |
> | XSTest-Safe | 0.39 / 0.55 | 0.48 / 0.50 | 0.71 |
> | OKTest | 0.41 / 0.55 | 0.44 / 0.44 | 0.60 |
>
> | Mistral-7B-v0.3 | Statement and Rationale (paper / varied) | Statement-Only (paper / varied) | Rationale-Only (paper) |
> | --- | --- | --- | --- |
> | Advbench | 0.01 / 0.10 | 0.05 / 0.05 | 0.04 |
> | Malicious | 0.0 / 0.09 | 0.01 / 0.01 | 0.06 |
> | XSTest-Safe | 0.49 / 0.63 | 0.56 / 0.51 | 0.75 |
> | OKTest | 0.55 / 0.63 | 0.53 / 0.55 | 0.77 |
>
> | Gemma-2-9B | Statement and Rationale (paper / varied) | Statement-Only (paper / varied) | Rationale-Only (paper) |
> | --- | --- | --- | --- |
> | Advbench | 0.01 / 0.06 | 0.01 / 0.02 | 0.0 |
> | Malicious | 0.03 / 0.05 | 0.0 / 0.0 | 0.03 |
> | XSTest-Safe | 0.44 / 0.44 | 0.50 / 0.42 | 0.68 |
> | OKTest | 0.51 / 0.52 | 0.56 / 0.51 | 0.69 |
>
> | Qwen2.5-7B | Statement and Rationale (paper / varied) | Statement-Only (paper / varied) | Rationale-Only (paper) |
> | --- | --- | --- | --- |
> | Advbench | 0.02 / 0.04 | 0.01 / 0.02 | 0.03 |
> | Malicious | 0.02 / 0.03 | 0.0 / 0.0 | 0.03 |
> | XSTest-Safe | 0.39 / 0.43 | 0.31 / 0.42 | 0.65 |
> | OKTest | 0.55 / 0.65 | 0.51 / 0.55 | 0.76 |
>
> Replacing all refusal statements in the dataset with these diverse variants resulted in overall similar or improved compliance on pseudo-harmful benchmarks compared to the original refusal statements. However, performance remained consistently **below that of the Rationale-Only condition**. This indicates that while stylistic variation may reduce anchoring to specific refusal tokens, the presence of refusal statement continues to induce elevated false refusals, further supporting our core claim regarding template-induced effects.

---

> ### Author Response · Authors · 2025-11-20
> **Response to Reviewer 513G (2/2)**
>
> **2. Integration with other safety alignment techniques.**
>
> > It would be interesting to see how rationale-only supervision integrates with other common alignment techniques.
> >
>
> Thank you for raising this insightful question. As noted in Section 7 of the paper, rationale-only supervision is complementary to existing mitigation methods, and combining them can yield additional benefits. In response to the reviewer’s suggestion, we further examined how it behaves when paired with safety mechanisms such as Refusal Tokens [2] and a post-hoc safety classifier (WILDGUARD) [3]. We applied these mechanisms in combination with rationale-only supervision using the Llama-3.1-8B model. As shown below, both approaches improve safety on harmful benchmarks, while also affecting compliance on pseudo-harmful inputs:
>
> | Llama-3.1-8B | Rationale-Only | + Refusal Tokens | + WILDGUARD |
> | --- | --- | --- | --- |
> | AdvBench | 0.02 | 0.01 | 0.0 |
> | Malicious | 0.06 | 0.03 | 0.01 |
> | XSTest-Safe | 0.71 | 0.52 | 0.70 |
> | OKTest | 0.60 | 0.54 | 0.57 |
>
> Both techniques lead to stronger safety on harmful benchmarks, but they also introduce additional false refusals:
>
> - **Refusal Tokens:** As reported in [2], without contrastive supervision, refusal-token training can cause false refusal. In our setting, we could not incorporate contrastive set because including contrast sets would require adding pseudo-harmful examples to training—data we intentionally keep unseen to preserve fair evaluation—so the method predictably increases false refusals here as well.
> - **WILDGUARD:** Consistent with the original paper[3], the classifier occasionally misclassifies pseudo-harmful inputs as unsafe, thereby slightly increasing false refusals.
>
> Nonetheless, this suggests a practical direction: in deployment, one could combine rationale-only supervision with safety mechanisms tuned to strike a desirable balance between safety guarantees and minimized false refusal.
>
> ---
> **references**
>
>
> [1] Xiangyu, Qi, et al. "Safety Alignment Should Be Made More Than Just a Few Tokens Deep." International Conference on Learning Representations (2025).
>
> [2] Jain, Neel, et al. "Refusal tokens: A simple way to calibrate refusals in large language models." arXiv preprint arXiv:2412.06748 (2024).
>
> [3] Han, Seungju, et al. "Wildguard: Open one-stop moderation tools for safety risks, jailbreaks, and refusals of llms." Advances in Neural Information Processing Systems 37 (2024): 8093-8131.

---

> > ### Author Response · Authors · 2025-11-26
> > **Gentle Reminder**
> >
> > **Dear Reviewer 513G,**
> >
> > We sincerely appreciate the time and effort you have devoted to reviewing our paper and providing valuable feedback. Your comments have been carefully considered, and we have endeavored to address all questions and concerns. It would be greatly appreciated if you could let us know whether our responses sufficiently clarify the points you raised.
> >
> > If any areas would benefit from further clarification or additional explanation, please do not hesitate to let us know. We would be happy to provide any further details you may require. Thank you once again for your thoughtful review and your time.

---

### Official Review · Reviewer_g3je · 2025-10-31

**Soundness:** 4
**Presentation:** 3
**Contribution:** 4
**Rating:** 8
**Confidence:** 3

**Summary:**

The paper studies where false refusals come from - when the model refuses innocuous requests that superficially resemble harmful ones. Their core finding is that by decomposing the model response to harmful requests into the generic refusal statement and the rationale and feeding just the rationale to the model during SFT, it's possible to reduce the false refusal rate. They also find that the position of the refusal statement matters: namely it's better to put it at the end. They also find that the more specific the rationale the better as far as reducing false refusals. They also analyze the reasons for this, finding that the model maintains higher entropy over token space and is able to distribute attribution across tokens in the prompt when tuned on rationales, whereas when trained on refusals the model tends to attribute to only a single token in the prompt.

**Strengths:**

The paper contributes to the understanding of where false refusals come from, and has clear implications for the design of safety tuning datasets. Namely model responses to harmful queries should include highly specific rationale for why the query cannot be fulfilled, and can omit a generic refusal statement without compromising safety. In addition, the paper offers a theory of why this is the case in terms of token entropy and attributions.

**Weaknesses:**

Would have been nice to see results for different model sizes within a family. Also would be have been nice to see how safety / refusal rates for the pretrained models after the experimental interventions in the paper compare to those of the released instruction-tuned variants.

nit line 376 should be Rationale-Only models, not Rationale-Onlymodels

**Questions:**

When comparing request-specific versus generic rationales, were the results (in table 2) averaged over all refusal statement positions? Or was that comparison only for rationale-only conditions

---

> ### Author Response · Authors · 2025-11-20
> **Response to Reviewer g3je**
>
> We sincerely thank the reviewer for the positive and thoughtful assessment of our work. We are grateful for the careful reading and the constructive feedback, which have been very helpful in improving the clarity and completeness of the paper. We hope that the additional analyses and clarifications below address the reviewer’s remaining questions.
>
> &nbsp;
>
> **1. Results for different model sizes.**
>
> > Would have been nice to see results for different model sizes within a family.
>
>
> Thank you for raising this helpful point. To directly address it, we extend our experiments to include both smaller and larger models within the same families (Gemma, Qwen2.5, Llama-3.1). Across all model families, we observe the consistent pattern:
>
> - Models trained with refusal statements (Statement-Only or Statement and Rationale) show higher false refusal rates.
> - Rationale-Only training reliably reduces false refusals while maintaining comparable safety.
>
> Below, we summarize the results (compliance rates) grouped by model sizes.
>
> **Smaller Models**
>
> | Gemma-2-2B | Statement-Only | Rationale-Only | Statement and Rationale |
> | --- | --- | --- | --- |
> | Advbench | 0.04 | 0.03 | 0.02 |
> | Malicious | 0.01 | 0.10 | 0.05 |
> | XSTest-Safe | 0.4 | 0.66 | 0.47 |
> | OKTest | 0.43 | 0.71 | 0.5 |
>
> | Qwen2.5-3B | Statement-Only | Rationale-Only | Statement and Rationale |
> | --- | --- | --- | --- |
> | Advbench | 0.02 | 0.01 | 0.02 |
> | Malicious | 0.0 | 0.07 | 0.02 |
> | XSTest-Safe | 0.36 | 0.72 | 0.47 |
> | OKTest | 0.43 | 0.66 | 0.49 |
>
> **Larger Models**
>
> | Llama-3.1-70B | Statement-Only | Rationale-Only | Statement and Rationale |
> | --- | --- | --- | --- |
> | Advbench | 0.0 | 0.01 | 0.02 |
> | Malicious | 0.0 | 0.02 | 0.02 |
> | XSTest-Safe | 0.38 | 0.75 | 0.50 |
> | OKTest | 0.42 | 0.68 | 0.53 |
>
> | Qwen2.5-72B | Statement-Only | Rationale-Only | Statement and Rationale |
> | --- | --- | --- | --- |
> | Advbench | 0.0 | 0.02 | 0.01 |
> | Malicious | 0.0 | 0.03 | 0.02 |
> | XSTest-Safe | 0.53 | 0.79 | 0.42 |
> | OKTest | 0.60 | 0.74 | 0.54 |
>
> Across both small and large models, and across all three families, the same behavior emerges: Refusal statements consistently increase false refusals, while rationale-only supervision reliably mitigates them even at substantially different model scales.
>
>
> &nbsp;
>
>
> **2. Comparison with instruction-tuned variants.**
>
> > Also would be have been nice to see how safety / refusal rates for the pretrained models after the experimental interventions in the paper compare to those of the released instruction-tuned variants.
> >
>
> We appreciate the reviewer’s thoughtful recommendation. We agree that comparing against released instruction-tuned variants can offer complementary insights. However,  such models are trained with undisclosed proprietary datasets and alignment pipelines, which differ substantially from our dataset-controlled setup and make direct interpretation of differences challenging. Nevertheless, following your suggestion, we evaluated  the instruction-tuned versions of the models used in this study on our benchmarks, and the resulting compliance rates are presented below.
>
> |  | Llama-3.1-8B-Instruct | Mistral-7B-Instruct-v0.3 | Gemma-2-9B-it | Qwen2.5-7B-Instruct |
> | --- | --- | --- | --- | --- |
> | Advbench | 0.07 | 0.69 | 0.03 | 0.13 |
> | Malicious | 0.04 | 0.78 | 0.04 | 0.25 |
> | XSTest-Safe | 0.85 | 0.96 | 0.76 | 0.93 |
> | OKTest | 0.82 | 0.84 | 0.72 | 0.88 |
>
> As the table shows, Mistral-7B-Instruct-v0.3 and Qwen2.5-7B-Instruct exhibit high compliance rate across both harmful and pseudo-harmful inputs. In contrast, Llama-3.1-8B-Instruct and Gemma-2-9B-it show more balanced behavior and still maintain relatively high compliance on pseudo-harmful benchmarks. These results illustrate how proprietary alignment pipelines can produce widely varying refusal behaviors across models, making direct comparison difficult.
>
> &nbsp;
>
>
> **3. Typos.**
>
> Thank you for reading so carefully and pointing this out; we will correct the typo to “Rationale-Only models”.
>
> &nbsp;
>
> **4. Results of request-specific versus generic rationales.**
>
> > When comparing request-specific versus generic rationales, were the results (in Table 2) averaged over all refusal statement positions? Or was that comparison only for rationale-only conditions
> >
>
> Thank you for the clarifying question. The request-specific and generic rationale conditions were constructed as two independent variants of the rationale-only setting, where the explanatory content is modified to either explicitly reference the user query or intentionally generalize it. These datasets do not depend on refusal-statement positions and are evaluated separately from those experiments.
>
> This comparison was designed to explore which aspects of rationales contribute to improved differentiation between harmful and pseudo-harmful inputs and to provide additional evidence for understanding the factors behind false refusals. We will clarify this distinction more explicitly in the revised version.

---

> > ### Author Response · Authors · 2025-11-26
> > **Gentle Reminder**
> >
> > **Dear Reviewer g3je,**
> >
> > We sincerely appreciate the time and effort you have devoted to reviewing our paper and providing valuable feedback. Your comments have been carefully considered, and we have endeavored to address all questions and concerns. It would be greatly appreciated if you could let us know whether our responses sufficiently clarify the points you raised.
> >
> > If any areas would benefit from further clarification or additional explanation, please do not hesitate to let us know. We would be happy to provide any further details you may require. Thank you once again for your thoughtful review and your time.

---

### Meta-Review · Area_Chair_oLmx · 2025-12-03

**Summary:**

Reviewers EfgL and 8bez argued the method is "too simple" characterizing it as a trivial data formatting change rather than a significant algorithmic contribution. Reviewer 8bez persistently argued that rationales might become new shortcuts themselves predicting that models might learn to blindly refuse any prompt containing words found in safety explanations (e.g., refusing "kill time" because "kill" appears in the rationale). Reviewers initially questioned if the findings would hold up on larger models (e.g., 70B parameters) or with diverse refusal styles beyond standard boilerplate, though the authors provided additional experiments to address this.

Overall, I do not often penalize simple papers. I think simplicity is key. However, for a NeurIPS of such high bar, I would expect much more from an experimental work. Particularly, understanding through theory or practice the reasons behind why the rational only works better at improving false refusals (better trading off utility with safety). For instance, mechanistic interpretability, SAE, probing, etc., could potentially be an addition to this paper to try and justify the reasons (or at least correlations) behind why this is the case. The experiments on first-token entropy are not particularly surprising. Well, if you do train on harmful data to generate the same response that is often I'm sorry with cross-entropy loss, you are indirectly (can be proven) minimizing the entropy of generating this conditioned on the harmful request. While this experiment is commendable, it is just saying we minimized a cross-entropy loss for this fixed target but worded differently, i.e., are there particular internal activations that are associated with explanation-related concepts that fire more compared to the non-refusal-only response?

Maybe some more experiments on counterfactuals? The authors could have trained a model on mismatched rationales (e.g., refusing a harmful prompt with a nonsense explanation or a safe explanation). If the model still refuses based on the structure of the explanation rather than its content, it would support the reviewer’s fear of a new shortcut. This might be helpful in addressing one of the reviewers’ complaints about shortcuts. What about injecting the rationales during fine-tuning with a specific set of words that are harmless or nonsensical?

What about multi-turn refusal, which is something I hardly see in papers (potentially t = 2; where once the model refuses from answering a harmful request it is prompted again with a convincing argument)? Would this proposed method be better than current methods because the model learns more semantics?

The paper is on the right track, but its simplicity means that to be accepted as valuable, the bar is high when it comes to demonstrating that this simple approach is in fact universally reliable.

**Reviewer Concerns:**

see above.

**Reviewer Scores:**

see above.

---

### Decision · Program_Chairs · 2026-01-26

Reject